# Deep Learning Models Based on Pretreatment MRI and Clinicopathological Data to Predict Responses to Neoadjuvant Systemic Therapy in Triple-Negative Breast Cancer

**DOI:** 10.3390/cancers17060966

**Published:** 2025-03-13

**Authors:** Zhan Xu, Zijian Zhou, Jong Bum Son, Haonan Feng, Beatriz E. Adrada, Tanya W. Moseley, Rosalind P. Candelaria, Mary S. Guirguis, Miral M. Patel, Gary J. Whitman, Jessica W. T. Leung, Huong T. C. Le-Petross, Rania M. Mohamed, Bikash Panthi, Deanna L. Lane, Huiqin Chen, Peng Wei, Debu Tripathy, Jennifer K. Litton, Vicente Valero, Lei Huo, Kelly K. Hunt, Anil Korkut, Alastair Thompson, Wei Yang, Clinton Yam, Gaiane M. Rauch, Jingfei Ma

**Affiliations:** 1Department of Imaging Physics, The University of Texas MD Anderson Cancer Center, 1515 Holcombe Blvd., Houston, TX 77030, USA; zxu5@mdanderson.org (Z.X.);; 2Department of Biostatistics, The University of Texas MD Anderson Cancer Center, 1515 Holcombe Blvd., Houston, TX 77030, USA; 3Department of Breast Imaging, The University of Texas MD Anderson Cancer Center, 1515 Holcombe Blvd., Houston, TX 77030, USA; 4Department of Breast Medical Oncology, The University of Texas MD Anderson Cancer Center, 1515 Holcombe Blvd., Houston, TX 77030, USA; 5Department of Pathology, The University of Texas MD Anderson Cancer Center, 1515 Holcombe Blvd., Houston, TX 77030, USA; 6Department of Breast Surgical Oncology, The University of Texas MD Anderson Cancer Center, 1515 Holcombe Blvd., Houston, TX 77030, USA; 7Department of Bioinformatics & Computational Biology, The University of Texas MD Anderson Cancer Center, 1515 Holcombe Blvd., Houston, TX 77030, USA; 8Section of Breast Surgery, Baylor College of Medicine, 7200 Cambridge St., Houston, TX 77030, USA; 9Department of Abdominal Imaging, The University of Texas MD Anderson Cancer Center, 1515 Holcombe Blvd., Houston, TX 77030, USA

**Keywords:** deep learning, pathological complete response, pretreatment prediction, transfer learning, triple-negative breast cancer

## Abstract

Deep learning models based on pretreatment MRI and clinicopathological data were constructed to predict responses to neoadjuvant systemic therapy in triple-negative breast cancer. The best-performing deep learning model developed from the pretreatment multiparametric breast MRI and clinicopathological data acquired from 282 patients with stage I–III triple-negative breast cancer enrolled in a prospective clinical trial achieved an AUC of 0.76 in the internal testing dataset. The deep learning model achieved an AUC of 0.72 in the external I-SPY 2 trial testing dataset. Tumor volume preprocessing affected the model performance; the 3D model frameworks, tumor ROI selection, and data inputs had minimal impact on the model performance.

## 1. Introduction

Triple-negative breast cancer (TNBC) is an aggressive subtype of breast cancer that accounts for approximately 20% of all breast cancers [1]. TNBC is a heterogeneous disease with diverse molecular subtypes and is typically treated with neoadjuvant systemic therapy (NAST) followed by surgery, although the response to NAST varies across these subtypes [2]. Rates of pathologic complete response (pCR) after NAST range from 20% to 64% [2]. Patients with TNBC treated with NAST who achieve a pCR have favorable disease-free and overall survival, while up to 80% of patients with residual disease after NAST will develop metastatic disease [3]. The recent approval of immunotherapy as NAST for TNBC has led to increased rates of pCR, at the expense of higher toxicity [4]. Therefore, the accurate early prediction of whether a patient will have a pCR to NAST is highly important. The development of early noninvasive biomarkers of response to NAST could enable personalized treatments and minimize toxicity by permitting the triage of responders to less aggressive treatment strategies [5] or treatment de-escalation, sparing nonresponders from ineffective treatment regimens [6].

Prior studies have integrated baseline and posttreatment data to identify tumor changes that predict treatment responses in TNBC. Some studies used morphological, kinetic, and diffusion parameters from dynamic contrast-enhanced MRI (DCE) and diffusion-weighted imaging (DWI) to predict pCR status via linear-regression-based analysis [7,8,9,10]. Several recent studies have improved prediction accuracy using deep learning (DL) models that incorporate MRI findings [11,12,13,14]. However, these DL-based studies relied on input data from both pre-NAST scans and follow-up scans acquired several months later, which potentially precluded the early triage of patients to alternative treatments.

To overcome this limitation, other recent studies have attempted to use only baseline data to predict pCR to NAST. One model [15] used baseline DCE images and clinical information from the I-SPY 1 trial and achieved a testing AUC of 0.85. Another model incorporated molecular information with imaging data, achieving an AUC of 0.83 in an independent testing dataset [16]. However, both models were trained and tested over all breast cancer subtypes. To the best of our knowledge, there are no reports on the use of DL models for pCR prediction based on baseline data exclusive for TNBC.

The goal of our study was to develop DL models for predicting pCR to NAST in TNBC patients using pretreatment multiparametric breast MRI and clinicopathological data. Our study utilized the largest dataset of TNBC patients, whereas all previously published studies have included an inhomogeneous cohort of all breast cancer subtypes. Further, our study used only the data that were available before the start of the neoadjuvant treatment, potentially enabling the more personalized clinical management of the patients at an earlier time point.

## 2. Materials and Methods

### 2.1. Patient Data

This prospective study was approved by the institutional review board of MD Anderson Cancer Center. The internal dataset was from patients with stage I–III TNBC enrolled in an institutional clinical trial [NCT02276443] in which the patients were prospectively monitored for a response to NAST.

A total of 299 patients with biopsy-confirmed TNBC were enrolled between April 2016 and September 2021. The inclusion criteria included baseline multiparametric MRI and clinicopathological data being available, the patient having received NAST, and the patient having undergone definitive surgery with the assessment of pCR status, with pCR defined as the absence of residual invasive disease in the breast and axilla. The exclusion criteria were inflammatory breast cancer; missing pCR status after surgery; the absence of multiparametric MRI (DCE or DWI) before NAST; incomplete clinical information; incomplete or uninterpretable MR images; or failed manual tumor segmentation due to technical issues. In total, 282 patients from the clinical trial were included in this study (Figure 1). These patients were randomly divided with a 5:1 ratio into a development (training and validation) dataset (*n* = 235) and an independent internal testing dataset (*n* = 47). The internal NAST included 4 cycles of dose-dense doxorubicin and cyclophosphamide followed by 12 cycles of paclitaxel or other agents. Data from these patients have been previously reported [13,17,18,19].

The external testing dataset was from the I-SPY 2 trial [20], in which patients received 12 weekly cycles of paclitaxel alone (standard of care) or with experimental agents followed by 4 cycles of doxorubicin and cyclophosphamide every 2 to 3 weeks [21]. A total of 985 patients, including 242 with TNBC, were enrolled during 2010–2016. Patients with cancer types other than TNBC were excluded, and the identical inclusion and exclusion criteria used in the internal trial were applied, resulting in 62 patients with TNBC in the external testing dataset (Figure 1).

### 2.2. Imaging and Clinicopathological Data

The image acquisition parameters for DCE and DWI are detailed in Table A1 in Appendix B. For DCE, the subtraction images (referred to as *DCE* hereafter) obtained by subtracting the precontrast images from the images acquired 2.5 min after contrast agent injection were used for development and testing. For DWI, only the b = 800 s/mm^2^ images (referred to as DWI hereafter) were used. The external image data were first resampled to the voxel size of the internal image data and then both datasets were processed with the same procedure thereafter. Semiautomatic segmentations with MIM Maestro 7.2.5 (MIM Software Inc., Cleveland, OH, USA), performed by two fellowship-trained breast radiologists [G.M.R. and R.M.M.] with 5 and 15 years of experience in breast MRI, respectively, were used as the reference masks for tumor voxels in both the DCE and DWI images for the internal dataset; the publicly available tumor segmentations for the I-SPY 2 dataset were used as the reference masks for the external testing dataset. Categorical clinicopathological data (clinical stage, T category, N category, stromal tumor-infiltrating lymphocyte level, Ki-67 index) were converted into numerical values. The image data were self-normalized at the subject level, while clinical information was normalized at the group level.

### 2.3. Models

In previous studies of MRI-based pCR prediction in TNBC, several variables were identified as potentially affecting model performance [11,13,14,16,22,23,24,25,26,27,28,29,30]. Therefore, we systematically investigated the pCR prediction performance of the models with those variables:(1)Two 3D frameworks for transfer learning: ResNet18 [22,23,26,31], with ~5 million parameters and ~1000 trainable parameters, and ResNeXt50 [16,24], with ~30 million parameters and ~4000 trainable parameters. These two frameworks were chosen for their popularity and reported success in applications of medical-imaging-based prediction/classification.(2)Tumor volume preprocessing: normalized tumor volume [11,13,25] and original tumor matrix size [14,16,26,27,28,29,30]. The normalization referred to the median tumor size across all 282 subjects in the internal dataset, defined as the median MRI-measured longest diameter (cm) in the baseline study, which was 2.8 cm. The histogram of the tumor volume normalization ratio is shown in Figure A1 in Appendix B. The purpose here was to evaluate if using the original or normalizing tumor volume in preprocessing would impact the model performance, since both approaches [11,13,14,16,25,26,27,28,29,30] are reported in the literature.(3)Tumor ROI selection (Figure 2, Figure A2 in Appendix B): ROI1, which included voxels within the aforementioned reference masks of tumors [11,16,27]; ROI2, which included voxels within a tight bounding box of the reference masks [13,14,25]; and ROI3, which included voxels within an enlarged bounding box of the ROI2 (dilated by 5 mm along the top, bottom, left, and right sides) [28,29,30].(4)Data inputs: DCE images only; DWI images only; DCE and DWI images; or DCE and DWI images plus clinicopathological information. For each input channel of the image, the base layer parameters of the chosen transfer-learned 3D framework were locked and then linked to the top layer, which remained unlocked for training. In models that fused multiple inputs, the base layer features of each input were combined through concatenation and then linked to the top layer.

Forty-eight prediction models were evaluated with the internal testing dataset. Thirty-six models were evaluated with the external testing dataset, since clinicopathological information was not available from the I-SPY 2 dataset for the development of all of the models. The model structures are illustrated in Figure 2, and the network configurations and software environment for model training are detailed in Appendix A. For models incorporating the DCE image series, the difference image—obtained by subtracting the precontrast image from the image acquired at 2.5 min after contrast agent injection—was used as the input. For models involving DWI, only the b = 800 s/mm^2^ data were used. The three voxel selection methods for model training are detailed under ‘Data Pre-processing’ and in Figure A2. Each of the 48 alternative models was cross-validated as described in ‘Model Development’. The receiver operating characteristic (ROC) curves of the top-performing model are presented in the ’Evaluation’ section and in Figure 3.

### 2.4. Statistical Analysis

Age, body mass index, and longest tumor diameter were summarized using their means and standard deviations; clinical stage, tumor category, nodal category, and NAST regimen were summarized using the number of subjects and percentages; and the stromal tumor-infiltrating lymphocyte level and Ki-67 index were summarized using their median and interquartile range.

Differences between the pCR and non-pCR groups in age, body mass index, and longest tumor diameter were assessed using a *t*-test; differences in clinical stage, tumor category, and nodal category were assessed using the Pearson χ^2^ test; and differences in stromal tumor-infiltrating lymphocytes and Ki-67 level were assessed using Fisher’s exact test [32]. Statistical analyses were carried out using R version 4.0.3 (R Foundation for Statistical Computing, Vienna, Austria). *p* values of less than 0.05 were considered statistically significant.

For each prediction model, performance was measured by the mean of the area under the receiver operating characteristic (ROC) curves (AUCs) from each of the five folds over the testing dataset. The AUCs were compared between models with Delong’s method.

## 3. Results

Patient demographics and baseline clinical information are presented in Table 1. The development dataset, internal testing dataset, and external testing dataset were composed of 235 patients (age 50.0 ± 11.7 years), 47 patients (age 49.2 ± 10.6 years), and 62 patients (age 48.8 ± 10.5 years), respectively. The longest tumor diameter was significantly smaller in patients with a pCR than in patients without a pCR in both the internal dataset (*p* < 0.001) and the external testing dataset (*p* = 0.017). In addition, in the internal dataset, stromal tumor-infiltrating lymphocyte level (*p* < 0.001) and Ki-67 index (*p* = 0.012) were higher in the pCR group than in the non-pCR group, and there was an association between pCR and a lower clinical stage (*p* = 0.018), T category (*p* < 0.001), and N category (*p* < 0.001). The percentages of pCR cases in the development dataset, internal testing dataset, and external testing dataset were 48% (113/235), 40% (19/47), and 45% (28/62), respectively.

### 3.1. Top-Performing Models

For the internal testing dataset, the top-performing DL model using only DCE images achieved a mean AUC of 0.69 (95% confidence interval [CI]: 0.53–0.84), which resulted from the combination of the original tumor volume, ROI2, and ResNet18 (Table 2, Figure A3 in Appendix B). The top-performing models using DWI images only and using both DCE and DWI images also resulted from the same combination, achieving mean AUCs of 0.72 (95% CI: 0.56–0.86) for DWI images only (Table 2, Figure A4 in Appendix B), and 0.73 (95% CI: 0.57–0.86) using both DCE and DWI images (Table 2, Figure A5 in Appendix B). The top-performing model using DCE images, DWI images, and clinicopathological data achieved a mean AUC of 0.76 (95% CI: 0.60–0.88), which resulted from the combination of the original tumor volume, ROI2, and ResNeXt50 (Table 2, Figure A6 in Appendix B).

The logistic regression model using only clinical information achieved a mean AUC of 0.61 (range: 0.59–0.67) in the internal testing dataset.

For the external testing dataset, the top-performing DL model using only DCE images achieved a mean AUC of 0.72 (95% CI: 0.57–0.84), which resulted from the combination of the original tumor volume, ROI3, and ResNeXt50 (Table 3, Figure A3 in Appendix B). The top-performing DL model using only DWI images also achieved a mean AUC of 0.72 (95% CI: 0.56–0.86), which resulted from the combination of the original tumor volume, ROI3, and ResNet18 (Table 3, Figure A4 in Appendix B). The top-performing DL model using both DCE and DWI images achieved a mean AUC of 0.71 (95% CI: 0.56–0.84), which also resulted from the combination of the original tumor volume, ROI3, and ResNet18 (Table 3, Figure A5 in Appendix B).

The top-performing models by data inputs for the internal testing dataset (four models) and external testing dataset (three models) are summarized in Figure 3, and the confidence interval of each model is plotted in Figure A3, Figure A4, Figure A5 and Figure A6. The best-performing model in the internal testing dataset achieved a mean AUC of 0.76 and used DCE images, DWI images, and clinicopathological data. The best-performing model in the external testing dataset achieved a mean AUC of 0.72 and used only DCE images and only DWI images, respectively.

### 3.2. Impact of 3D Frameworks, Tumor Volume Preprocessing, and Tumor ROI Selection

The 3D frameworks did not affect model performance. Across all paired combinations, model performance did not differ significantly between ResNet18 and ResNeXt50. Among the seven top-performing models, five were composed with ResNet18, and two were composed with ResNeXt50.

Tumor volume preprocessing had a major impact on model performance. In models with paired combinations, using the original tumor volume resulted in a higher mean AUC than using the normalized tumor volume. For example, across models using DCE images only, eight (two with internal testing dataset, six with external testing dataset) of the twelve paired comparisons achieved higher mean AUCs with the original tumor volume than with the normalized tumor volume.

Tumor ROI selection did not affect model performances. In the seven top-performing models, replacing the selected ROI with one of the other two ROI options in the paired combination resulted in a difference in the mean AUC of at most 0.04.

## 4. Discussion

In this study, we developed and validated 3D DL models based on pretreatment data for predicting pCR in TNBC patients undergoing NAST. The best-performing model in the internal testing dataset included DCE images, DWI images, and clinicopathological data and achieved a mean AUC of 0.76. The best-performing model in the external testing dataset included only DCE images or DWI images and achieved a similar mean AUC of 0.72. Furthermore, we found that tumor volume preprocessing affected model performance, but 3D model frameworks, tumor ROI selection, and data inputs had minimal impact on model performance.

Several previous studies used only pretreatment data to predict pCR in breast cancer patients [14,16,28,33]. In a study including 356 patients (unknown number with TNBC) with different breast cancer molecular subtypes, models using DCE and molecular information were developed [16]. The best-performing DL model used the image-kinetic of DCE and molecular information and achieved an AUC of 0.83, while the model directly using DCE images and molecular data achieved an AUC of 0.79. In another study using the pretreatment data of 536 breast cancer patients (unknown number with TNBC), in which 429 patients were used for training and 107 were used for testing, the best-performing DL model used T1-weighted images, T2-weighted images, and clinical information and achieved an AUC of 0.89 [28]. The model using only T1-weighted images achieved an AUC of 0.73, the model using only T2-weighted images achieved an AUC of 0.66, and the model using clinical information alone achieved an AUC of 0.83. In contrast to prior reports, which were based on retrospective data analysis, our study used a well-curated prospectively collected dataset including only TNBC patients. Our top-performing models using only DCE images achieved AUCs of 0.69 for the internal testing dataset and 0.72 for the external testing dataset, which are comparable to the AUCs in the models from prior reports under similar settings.

Similarly to previous studies that showed the superior performance of models with multiple inputs over models with single inputs [9,16,31,32], our study showed that the best-performing model used DCE, DWI, and clinicopathological information. Among the models using DCE images alone or DWI images alone, we observed a large variation in AUC values in both the internal and external testing dataset. A logistic regression model using only clinicopathological information also performed poorly.

This study differs from previous studies in the investigation of input variables that affect model performance. We found that ResNet18, with its lower computational demands and faster performance in both model development and testing, may be better suited than ResNeXt50 for clinical applications. Unlike previous studies [11,13,25] that normalized tumor volume during preprocessing, we found that the original tumor volume without normalization was satisfactory for our model performance. Interestingly, we found that tumor ROI selection did not affect the model performance. Previous studies [11,13,14,16,25,27,28,29,30] have reported a good prediction performance with either accurate segmentation or bounding boxes, consistent with our findings. The lack of influence of the tumor ROI selection method on model performance suggests that our models can be used without often labor-intensive manual tumor segmentation.

The patients in our internal testing dataset received different NAST regimens to the ones used for patients in the external testing dataset. However, our top-performing model achieved a similar and even slightly better performance in the external testing dataset, suggesting that our model is stable across NAST regimens.

This study has several limitations. First, our DL models were developed only with DCE images, DWI images, and clinical information. Including other imaging modalities [11] (such as T2W images) or features [17,18,19] (such as radiomic information) might further improve prediction accuracy. A recent study reported that a DL model built from pathological slides acquired during the baseline biopsy of 540 breast cancer patients achieved an AUC of 0.85 [34]. Our results may be improved upon with the addition of digitized pathological information. Second, the methods used to integrate multiple inputs need further investigation. Our DL framework fused different inputs by concatenating the top-layer elements of each input at the “feature level”. Although this is a well-accepted data fusion strategy for classification purposes, some investigators [35,36] have also proposed fusion at the “decision level”, which is independent of the disparity between the original data dimensions among the modalities. Finally, the current models need to be validated on a larger dataset to improve their statistical confidence and reliability. We were not able to test our model based on DCE images, DWI images, and clinicopathological data on the external testing dataset due to insufficient available clinicopathological information. The comparable AUCs in the internal and external testing datasets demonstrated the reliability of our model across various NAST regimens. We also would expect a smaller confidence interval at a larger testing dataset.

## 5. Conclusions

In conclusion, we were able to develop 3D DL models for predicting pCR to NAST in TNBC based on multiparametric MR images and clinical information acquired before the initiation of therapy. The model integrating DCE, DWI, and clinical information provided the best prediction performance. This model could be used for the triaging of TNBC patients to the most successful and least toxic treatment regimens while achieving the best outcomes.

## Figures and Tables

**Figure 1 cancers-17-00966-f001:**
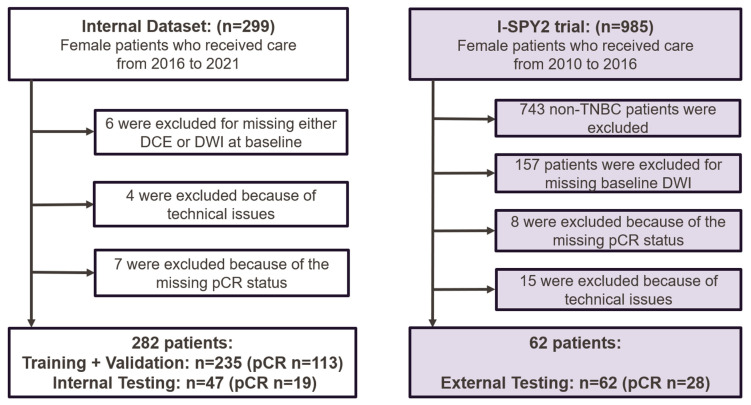
Flowcharts of patient recruitment and exclusion for the internal and external datasets. The external data (I-SPY 2 Imaging Cohort 1) are publicly available and were downloaded from The Cancer Imaging Archive. DCE = dynamic contrast-enhanced MRI; DWI = diffusion-weighted imaging; pCR = pathologic complete response; TNBC = triple-negative breast cancer.

**Figure 2 cancers-17-00966-f002:**
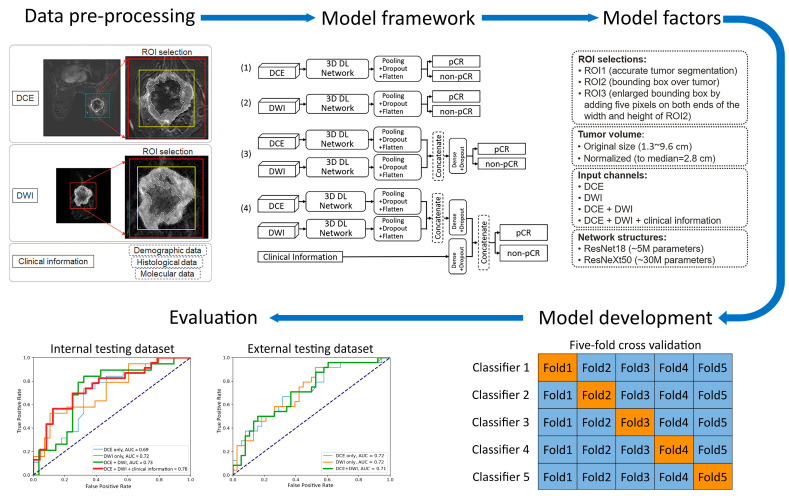
Workflow of training the deep learning (DL) prediction models. In “Model development”, training folds are indicated with blue and validation folds are indicated with orange. AUC = area under the ROC curve; pCR = pathologic complete response; ROI = region of interest; and TNBC = triple-negative breast cancer.

**Figure 3 cancers-17-00966-f003:**
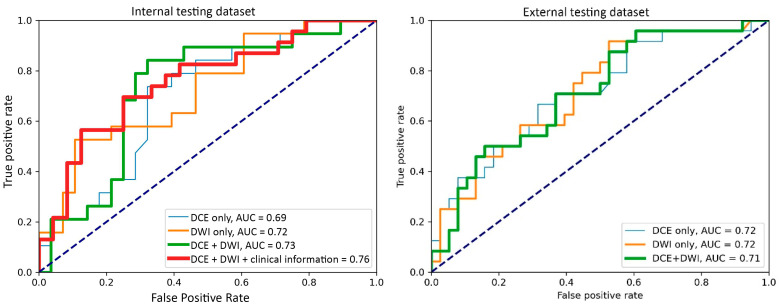
Receiver operating characteristic curves of the best-performing models with the input of the dynamic contrast-enhanced MRI (DCE) images only, the diffusion-weighted imaging (DWI) images only, DCE with DWI images, and DCE and DWI with clinicopathological information.

**Table 1 cancers-17-00966-t001:** Demographics and baseline clinical information of patients in the (A) internal and (B) external dataset.

**(A)**				
**Internal Data**	**All Patients**	**pCR**	**non-pCR**	** *p* ** **Value**
No. of patients	282	132	150	
Age, mean ± SD, y	49.8 ± 11.4	49.5 ± 11.5	50.1 ± 11.5	0.36
BMI, mean, kg/m^2^	28.9	28.5	29.2	0.15
Longest tumor diameter, mean ± SD, cm	3.4 ± 1.5	2.9 ± 1.3	3.6 ± 1.8	<0.001
Clinical stage, n (%)				0.018
I	34 (12)	19 (14)	15 (10)	
II	206 (73)	100 (76)	106 (71)	
III	42 (15)	13 (10)	29 (19)	
T category, n (%)				<0.001
T1	51 (18)	31 (23)	20 (13)	
T2	189 (67)	91 (69)	98 (65)	
T3	36 (13)	8 (6)	28 (19)	
T4	6 (2)	2 (2)	4 (3)	
N category, n (%)				<0.001
N0	181 (64)	92 (70)	89 (59)	
N1	72 (26)	30 (23)	42 (28)	
N2	7 (2)	3 (2)	4 (3)	
N3	22 (8)	7 (5)	15 (10)	
Stromal tumor-infiltrating lymphocyte level, %, median (IQR)	10 (4–20)	20 (4–30)	10 (4–20)	<0.001
Ki-67 index, %, median (IQR)	70 (50–90)	75 (50–90)	70 (51–87)	0.012
NAST regimen, n				
Doxorubicin + Paclitaxel	233	121	112	
Doxorubicin + Enzalutamide	11	3	8	
Doxorubicin + Panitumumab	15	2	13	
Doxorubicin + Everolimus	7	0	7	
Doxorubicin + Atezolizumab	12	5	7	
Doxorubicin + Alpelisib	4	1	3	
**(B)**				
**External Data**	**All Patients**	**pCR**	**non-pCR**	** *p* ** **Value**
No. of patients	62	28	34	
Age, mean ± SD, y	48.8 ± 10.5	49.4 ± 10.4	48.2 ± 10.9	0.62
Longest tumor diameter, mean ± SD, cm	4.1 ± 2.3	3.5 ± 1.1	4.7 ± 2.9	0.017
T category, n (%)				0.11
T2	5 (8)	1 (4)	4 (12)	
T3	56 (90)	26 (92)	30 (88)	
N/A	1 (2)	1 (4)	0 (0)	
NAST regimen, n				
Paclitaxel	13	3	10	
Paclitaxel + MK-2206	9	6	3	
Paclitaxel + AMG 386	15	7	8	
Paclitaxel + Ganetespib	1	0	1	
Paclitaxel + Ganitumab	24	12	12	

Note—BMI = body mass index; IQR = interquartile range; NAST = neoadjuvant systemic therapy; pCR = pathologic complete response; SD = standard deviation; and N/A = not available.

**Table 2 cancers-17-00966-t002:** AUCs by data inputs and other variables in the internal testing dataset.

Internal Testing Dataset						
Network	ResNet18			ResNeXt50		
	ROI1	ROI2	ROI3	ROI1	ROI2	ROI3
DCE-only models						
Normalized tumor volume	0.63	0.61	0.63	0.52	0.48	0.48
	(0.58, 0.7)	(0.59, 0.65)	(0.55, 0.68)	(0.44, 0.61)	(0.39, 0.62)	(0.45, 0.53)
Original tumor volume	0.68	0.69	0.68	0.66	0.65 *	0.65 *
	(0.62, 0.7)	(0.66, 0.74)	(0.66, 0.7)	(0.65, 0.67)	(0.64, 0.66)	(0.65, 0.66)
DWI-only models						
Normalized tumor volume	0.60	0.59	0.61	0.61	0.61	0.57
	(0.55, 0.68)	(0.49, 0.67)	(0.46, 0.74)	(0.61, 0.62)	(0.59, 0.63)	(0.56, 0.59)
Original tumor volume	0.69	0.72	0.70	0.53	0.58	0.60
	(0.67, 0.72)	(0.69, 0.75)	(0.69, 0.71)	(0.51, 0.56)	(0.53, 0.73)	(0.54, 0.73)
DCE + DWI models						
Normalized tumor volume	0.57	0.60	0.59	0.58	0.59	0.57
	(0.5, 0.63)	(0.59, 0.64)	(0.54, 0.62)	(0.54, 0.61)	(0.58, 0.6)	(0.56, 0.58)
Original tumor volume	0.71	0.73	0.72	0.71	0.70	0.71
	(0.67, 0.76)	(0.71, 0.74)	(0.68, 0.76)	(0.66, 0.78)	(0.65, 0.78)	(0.67, 0.77)
DCE + DWI + clinical information models						
Normalized tumor volume	0.68	0.69	0.69	0.67	0.68	0.67
	(0.66, 0.72)	(0.66, 0.72)	(0.66, 0.73)	(0.64, 0.70)	(0.65, 0.69)	(0.63, 0.70)
Original tumor volume	0.74	0.73	0.71	0.72	0.76	0.72
	(0.67, 0.79)	(0.67, 0.81)	(0.65, 0.76)	(0.69, 0.75)	(0.72, 0.78)	(0.68, 0.75)

Note—Values in table are mean (minimum, maximum) areas under the receiver operating characteristic curve (AUCs). The top-performing models are highlighted in bold. DCE = dynamic contrast-enhanced MR images; DWI = diffusion-weighted imaging images; ROI1 = accurate tumor segmentation; ROI2 = bounding box over tumor; and ROI3 = enlarged bounding box dilated by 5 mm along the top, bottom, left, and right sides of ROI2. * Higher AUC than the paired model with the other tumor volume preprocessing method, *p* < 0.05.

**Table 3 cancers-17-00966-t003:** AUCs by data inputs and other variables in the external testing dataset.

External Testing Dataset						
Network	ResNet18			ResNeXt50		
	ROI1	ROI2	ROI3	ROI1	ROI2	ROI3
**DCE-only models**						
Normalized tumor volume	0.41	0.46	0.53	0.43	0.44	0.44
	(0.33, 0.48)	(0.43, 0.57)	(0.49, 0.54)	(0.40, 0.53)	(0.40, 0.58)	(0.42, 0.53)
Original tumor volume	0.69 *	0.67 *	0.67 *	0.71 *	0.71 *	0.72 *
	(0.63, 0.69)	(0.64, 0.69)	(0.64, 0.69)	(0.71, 0.72)	(0.71, 0.72)	(0.71, 0.72)
**DWI-only models**						
Normalized tumor volume	0.61	0.57	0.68	0.53	0.62	0.61
	(0.52, 0.58)	(0.45, 0.63)	(0.59, 0.76)	(0.51, 0.54)	(0.60, 0.68)	(0.57, 0.71)
Original tumor volume	0.66	0.70	0.72	0.66	0.68	0.69
	(0.65, 0.67)	(0.68, 0.73)	(0.71, 0.72)	(0.64, 0.66)	(0.60, 0.72)	(0.68, 0.69)
**DCE + DWI models**						
Normalized tumor volume	0.62	0.61	0.68	0.59	0.63	0.64
	(0.49, 0.60)	(0.38, 0.62)	(0.57, 0.68)	(0.52, 0.65)	(0.63, 0.65)	(0.62, 0.65)
Original tumor volume	0.67	0.70	0.71	0.66	0.67	0.71
	(0.64, 0.70)	(0.69, 0.72)	(0.67, 0.73)	(0.55, 0.68)	(0.50, 0.70)	(0.60, 0.72)

Note—Values in the table are the mean (minimum, maximum) areas under the receiver operating characteristic curve (AUCs). The top-performing models are highlighted in bold. DCE = dynamic contrast-enhanced MR images; DWI = diffusion-weighted imaging images; ROI1 = accurate tumor segmentation; ROI2 = bounding box over tumor; and ROI3 = enlarged bounding box dilated by 5 mm along the top, bottom, left, and right sides of ROI2. * Higher AUC than the paired model with the other tumor volume preprocessing method, *p* < 0.05.

## Data Availability

Data are available from the corresponding author upon request.

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
