# Peer review of "Deep Learning Models Based on Pretreatment MRI and Clinicopathological Data to Predict Responses to Neoadjuvant Systemic Therapy in Triple-Negative Breast Cancer"

_cancers, 2025, doi:10.3390/cancers17060966_

Round 1
Reviewer 1 Report
Comments and Suggestions for Authors
This is a deep-learning-based study for the prediction of pathologic complete response to neoadjuvant systemic therapy in patients with TNBC using baseline multiparametric MRI and clinicopathological data. The top-performing model, integrating DCE and DWI with clinicopathological features, achieved an AUC of 0.76 in internal testing and 0.72 in external validation using the I-SPY 2 dataset. This study further elucidates early noninvasive biomarkers of treatment response, guiding personalized therapeutic strategy.
Strengths:
Systematic comparison of models: Running the comparison for 48 different models, varying architecture, tumor segmentation strategy, and imaging input, strengthens generalizability of results.
Major Limitations and Comments:
- Clinical Relevance: This study should focus more on clinical relevance and applicability of their model. For example, helping in reducing the need for unnecessary mastectomies after pCR. Please cite this study PMID: 39362047 to improve your study and the clinical relevance of it.
- Modest Predictive Performance: The best overall model performance had an AUC of 0.76, which is nonetheless relatively poor compared to other biomarker-informed predictive models, including molecular or radiomics-based ones.
Authors should discuss how additional imaging modalities, such as T2-weighted MRI or radiomics features, or genomic biomarkers might improve performance.
- Lacked the Clinical Model External Validation: Since the I-SPY 2 dataset had missed the clinicopathologic details, this model which includes DCE, DWI, and clinicopathological data could not get external validated.
This is a concern over the overfitting to the internal dataset, which calls for further independent validation in larger cohorts.
- NAST Regimens Lacked Detailed Analysis: The internal and external datasets treated the patients with varied chemotherapy regimens, although model performance stratification by type of treatment was not given.
Since both immunotherapy and novel agents greatly influence pCR rates, subgroup analyses are necessary to test model reliability across these treatment variations.
- Interpretability of Deep Learning Models: Lack of explanation techniques such as SHAP or Grad-CAM in the study to interpret decisions of DL models.
This would provide better clinical trust and applicability with visual heatmaps or feature importance rankings.
Author Response
To Reviewer 1:
We appreciate your detailed comments and suggestions regarding data validation and its application value. Please find our responses below and the revisions highlighted with Track Changes in the re-submitted files.
- Clinical Relevance: This study should focus more on clinical relevance and applicability of their model. For example, helping in reducing the need for unnecessary mastectomies after pCR. Please cite this study PMID: 39362047 to improve your study and the clinical relevance of it.
We agree with the reviewer that accurate pCR prediction could potentially help to avoid mastectomies for some patients with pCR. The major goal of our study, however, is to predict pCR using the baseline imaging and clinical information so more personalized decisions can be made early during their neoadjuvant treatment. We have added the suggested reference as Ref #5 in the revision.
- Modest Predictive Performance: The best overall model performance had an AUC of 0.76, which is nonetheless relatively poor compared to other biomarker-informed predictive models, including molecular or radiomics-based ones.
We agree with the reviewer that the AUC of our prediction model is modest and not as high as some that have been reported in the literature in the other contexts. We note again that our model relies on only the baseline pretreatment imaging or clinical information. Further, we note that our model AUC are validated on both internal and external independent datasets. Additional improvements may be achieved by incorporating other pretreatment images and/or genomic information that are available.
- Authors should discuss how additional imaging modalities, such as T2-weighted MRI or radiomics features, or genomic biomarkers might improve performance.
Our group have investigated using radiomics analyses for pCR prediction and the findings are reported in Ref. #19 [PMID:38992094]. The genomic information can be incorporated into our model but was not available at the time of this study.
- Lacked the Clinical Model External Validation: Since the I-SPY 2 dataset had missed the clinicopathologic details, this model which includes DCE, DWI, and clinicopathological data could not get external validated.
As the reviewer rightfully pointed out, our best performing model that includes DCE, DWI, and clinicopathological data was evaluated only on our internal independent testing dataset. We performed an extensive search of all the publicly available databases and did not find any on breast cancer that contain DCE, DWI, and clinicopathological data. Nonetheless, we were able to test our model that includes DCE and DWI on the external independent I-SPY2 dataset with a stable prediction performance.
- This is a concern over the overfitting to the internal dataset, which calls for further independent validation in larger cohorts.
Although further independent validation in larger cohorts will always be helpful, our models are developed and tested with the largest cohort of triple negative breast cancer that we are aware of. Furthermore, our model that includes only imaging data was tested on a larger external testing dataset from I-SPY2 than the internal testing dataset, with comparable AUCs. We believe these results alleviate the concern of overfitting in our models.
- NAST Regimens Lacked Detailed Analysis: The internal and external datasets treated the patients with varied chemotherapy regimens, although model performance stratification by type of treatment was not given.
Although using only the patients with the same disease (triple negative breast cancer) and undergoing identical treatment regimens is hypothetically a “cleaner” study design, we feel it is neither practical nor necessary. There will always be variations in the real-world clinical practice. Most of the patients in our internal dataset (233 out of 282 subjects) received the 'Doxorubicin + Paclitaxel' regimen, while the remaining five regimens were applied to an average of only 10 subjects. Similarly, the external dataset included an average of 12 subjects per regimen. Performing model development and testing according to the treatment regimens would require an enormously larger patient enrollment.
- Since both immunotherapy and novel agents greatly influence pCR rates, subgroup analyses are necessary to test model reliability across these treatment variations.
We agree with the reviewer and point out that our study was performed on the patients receiving the conventional neoadjuvant treatment, before immunotherapy was approved by FDA. We intend to expand our models to the patients receiving the immunotherapy. A preliminary account of our work was just accepted by and will be presented at the upcoming 2025 Annual Scientific Meeting of the International Society of Magnetic Resonance in Medicine.
- Interpretability of Deep Learning Models: Lack of explanation techniques such as SHAP or Grad-CAM in the study to interpret decisions of DL models. This would provide better clinical trust and applicability with visual heatmaps or feature importance rankings.
We acknowledge that applying techniques such as SHAP or Grad-CAM may enhance the transparency of a model’s decision-making process. however, these methods primarily provide interpretability at an individual case level, whereas our focus is on constructing a prediction model to extract features at the group level. We have provided performance metrics for each model and a controlled comparison of each factor to support the robustness of our model’s prediction. Being able to examine the model interpretability for clinical trust and applicability on an individual basis is an interesting question and may require a separate study in the future.
Reviewer 2 Report
Comments and Suggestions for Authors
I have reviewed your study titled "Deep learning models based on pretreatment MRI and clinicopathological data to predict response to neoadjuvant systemic therapy in triple-negative breast cancer" in detail. I have listed the points I found lacking in the article. I think that if these deficiencies are eliminated, the article will be at a better point. A short paragraph about the article's purpose is written in the last paragraph of the introduction. This paragraph should be expanded. Why the study was conducted, its innovative aspects, its contributions to the literature, etc. should be added. There are many studies conducted on the subject. However, no literature review was conducted in the article. The results were obtained with resnet-based models in the study and these results were not compared with other models or literature. The introduction section should be concluded with a paragraph that includes the organization of the article. The figure descriptions are written very long in some of them. It would be a more accurate approach to present this as a paragraph. The study should include why resnet-based models are used when many popular models exist in the literature. I think the performance metrics are low. To prove that these values ​​are high, you must include a literature review and the performances of other models and prominently highlight the study's contributions to the literature. The study's limitations should be detailed, and the conclusion section should be expanded.
Author Response
We appreciate your detailed comments and suggestions regarding data validation and its application value. Please find our responses below and the revisions highlighted with Track Changes in the re-submitted files.
- A short paragraph about the article's purpose is written in the last paragraph of the introduction. This paragraph should be expanded. Why the study was conducted, its innovative aspects, its contributions to the literature, etc. should be added. There are many studies conducted on the subject. However, no literature review was conducted in the article.
We appreciate the suggestion and have expanded the last paragraph of Introduction.
- The results were obtained with resnet-based models in the study and these results were not compared with other models or literature. The introduction section should be concluded with a paragraph that includes the organization of the article.
We chose the ResNet-based models because of their widespread use and success in medical imaging-based prediction and classification. In the revised manuscript, we have listed and reorganized the relevant references in the Methods Section (2.3.1).
- The figure descriptions are written very long in some of them. It would be a more accurate approach to present this as a paragraph. The study should include why resnet-based models are used when many popular models exist in the literature. I think the performance metrics are low. To prove that these values ​​are high, you must include a literature review and the performances of other models and prominently highlight the study's contributions to the literature.
Thank for the suggestion. We have reorganized the caption for Figure 2 and moved the details to the Methods Section (2.3). As indicated in the comment above, our choice of ResNet was based on its reported success for medical imaging related predictions. Potential performance improvements and future directions are added as a limitation in the Discussions section.
- The study's limitations should be detailed, and the conclusion section should be expanded.
Thank you for the suggestion. We further expanded the limitation and conclusion sections in the revision.
Reviewer 3 Report
Comments and Suggestions for Authors
The paper by Xu et al describes application of DL models for the classification of breast cancer based on MRI images and patient data analysis. Theme of the study is important and unteresting to a wide range of readers. The text is easy to read and contains all required information. The authors provided data about models architecture, training process and achieved results. The main findings are described at the text while details of the study are presented in the supplementary sections; such presentation of material is very comfortable for the readers. Thus, the paper leaves positive impression. So I recommend to publish this paper after correction of minor issues:
- Please provide CI values in figure 3 as well as in figuress B3-6.
- Number of cited references is quite limited. I propose to add about 15 references to the studies in the field. Such references may be added in the discussion section.
- Couclusion is very limited and too general. Please add exact obtained values and some ideas about further implementation of the proposed method into clinical practice.
Author Response
We appreciate your detailed comments and suggestions regarding data validation and its application value. Please find our responses below and the revisions highlighted with Track Changes in the re-submitted files.
- Please provide CI values in figure 3 as well as in figures B3-6.
Figure 3 summarizes B3 and B6. Including the confidence intervals (CIs) for each model within the figure would make the figure very busy and difficult to read. In the revised manuscript, we clarified that the CIs for each model in Figure 3 can be found in the appendix figures.
- Number of cited references is quite limited. I propose to add about 15 references to the studies in the field. Such references may be added in the discussion section.
Thank you for the suggestion. We have incorporated the reference suggested by Reviewer 1 to enhance the clinical relevance of our study. The revised manuscript now includes a total of 36 references, which is comparable to most publications in Cancers.
- Conclusion is very limited and too general. Please add exact obtained values and some ideas about further implementation of the proposed method into clinical practice.
Thank you for the suggestion. We expanded the limitation and conclusion section in the revised manuscript as suggested.
Round 2
Reviewer 1 Report
Comments and Suggestions for Authors
The manuscript can be accepted in the present form
Reviewer 2 Report
Comments and Suggestions for Authors
Congratulations on the successful revision.